# Incidental findings on brain imaging and blood tests: results from the first phase of Insight 46, a prospective observational substudy of the 1946 British birth cohort

Sarah E Keuss,[1] Thomas D Parker,[1] Christopher A Lane,[1] Chandrashekar Hoskote,[2] Sachit Shah,[2] David M Cash,[1] Ashvini Keshavan,[1] Sarah M Buchanan,[1] Heidi Murray-Smith,[1] Andrew Wong,[3] Sarah-Naomi James,[3] Kirsty Lu,[1] Jessica Collins,[1] Daniel G Beasley,[4] Ian B Malone,[1] David L Thomas,[5,6] Anna Barnes,[7] Marcus Richards,[3] Nick Fox,[1] Jonathan M Schott[1]

For numbered affiliations see end of article.

**Correspondence to**
Professor Jonathan M Schott;
j.schott@ucl.ac.uk

## ABSTRACT

**Objective** To summarise the incidental findings detected on brain imaging and blood tests during the first wave of data collection for the Insight 46 study.

**Design** Prospective observational sub-study of a birth cohort.

**Setting** Single-day assessment at a research centre in London, UK.

**Participants** 502 individuals were recruited from the MRC National Survey of Health and Development (NSHD), the 1946 British birth cohort, based on pre-specified eligibility criteria; mean age was 70.7 (SD: 0.7) and 49% were female.

**Outcome measures** Data regarding the number and types of incidental findings were summarised as counts and percentages, and 95% confidence intervals were calculated.

**Results** 93.8% of participants completed a brain scan (n=471); 4.5% of scanned participants had a pre-defined reportable abnormality on brain MRI (n=21); suspected vascular malformations and suspected intracranial mass lesions were present in 1.9% (n=9) and 1.5% (n=7) respectively; suspected cerebral aneurysms were the single most common vascular abnormality, affecting 1.1% of participants (n=5), and suspected meningiomas were the most common intracranial lesion, affecting 0.6% of participants (n=3); 34.6% of participants had at least one abnormality on clinical blood tests (n=169), but few reached the prespecified threshold for urgent action (n=11).

**Conclusions** In older adults, aged 69-71 years, potentially serious brain MRI findings were detected in around 5% of participants, and clinical blood test abnormalities were present in around one third of participants. Knowledge of the expected prevalence of incidental findings in the general population at this age is useful in both research and clinical settings.

## INTRODUCTION

Incidental clinical findings are often discovered during the course of conducting research. An incidental finding can be defined as 'a

### Strengths and limitations of this study

► A large number of participants underwent brain imaging and blood testing, at an almost identical age, and received feedback of incidental findings according to a prespecified standardised protocol.
► Participants were recruited from the 1946 British birth cohort, a broadly representative sample of the population born in mainland Britain during one week in 1946.
► Participant perception regarding the disclosure of incidental findings was not formally assessed, nor was the impact on their longer term health and psychological well-being.

finding concerning an individual research participant that has potential health or reproductive importance…but is beyond the aims of the study'.[1] The primary aim of most research is to generate data and advance knowledge, rather than to diagnose health problems in participants, and there is currently no legal requirement for researchers in the UK to report incidental findings to participants.[2] There are, however, important ethical reasons for disclosing certain incidental findings to participants in appropriate circumstances, particularly when they relate to serious and potentially treatable conditions.[1] It is therefore important that studies have protocols in place for managing them. While there is no consensus on how this should be done, it is recommended that researchers weigh up the potential benefits and harm to participants of being informed, as well as considering the associated time and cost, both to the study and to publicly funded health services.[2]

Incidental findings often lead to anxiety and have the potential to lead to unnecessary and invasive procedures for study participants.[3–5] Knowledge of the expected prevalence of incidental findings, based on clearly defined protocols for their determination, is important, allowing researchers to be better prepared for managing them and enabling study participants to be appropriately informed as part of the consent process. Given the increasing use of neuroimaging in primary, secondary and tertiary care, such information is also useful in the clinical setting, where it can facilitate management decisions. For example, knowing the probability of detecting an abnormality unrelated to a patient's symptoms might influence a clinician's decision to recommend a brain scan in a patient presenting with a benign-sounding headache, or prompt discussion with the patient regarding the pros and cons of scanning.

The MRC National Survey of Health and Development (NSHD) recruited 5362 individuals born in England, Scotland and Wales during the same week in 1946, and has followed them since birth, with over 2500 participants remaining in active follow-up.[6] Insight 46 is a longitudinal neuroimaging substudy of 502 NSHD participants, which aims to investigate genetic and life course factors that contribute to healthy and pathological brain ageing, in particular cerebrovascular and Alzheimer's disease. It involves detailed clinical phenotyping, brain magnetic resonance imaging (MRI), cerebral ß-amyloid positron emission tomography (PET), and blood and urine collection, at two time points approximately two years apart. The full study protocol, which includes clear criteria for reporting incidental findings, has been described elsewhere.[7]

The aim of this study is to summarise the incidental findings detected on brain MRI and blood tests during the first wave of data collection for Insight 46. Several studies have reported rates of incidental findings in different samples previously,[8 9] but to our knowledge none have reported on findings from a representative country-wide birth cohort.

## METHODS

### Recruitment

Individuals were recruited from NSHD participants who attended a study visit at age 60–64, who had previously indicated that they would be willing to consider participating in a study visit in London, and for whom relevant life course data were available (online supplementary file 1). NSHD participants who met these criteria were sent an information booklet about the study and then recruited by a study doctor via telephone. Those with known contraindications to PET or MRI scanning were not recruited. Eligibility criteria were relaxed towards the end of the study, allowing inclusion of some individuals with a few missing life course data points, in order to achieve the study's recruitment target.

### Consent

The booklet sent to participants prior to their visit contained a detailed description of the study tests, including information about the study protocol with regard to incidental findings. Specifically, it stated that 'we will inform you and your GP if any of the routine blood tests show any significant abnormalities' and we 'will let you and your doctor know if there are any major abnormalities on the MRI scan (eg, the presence of a tumour or a large aneurysm) which might affect your clinical care'. It also emphasised that 'being in a research study does not take the place of routine physical examinations or other appointments with your doctor and should not be relied upon to diagnose or treat medical problems'. All participants provided written consent to participate (online supplementary file 2). Prior to collecting blood samples, the study doctor asked participants whether they wished to opt out of receiving a copy of their blood results. This option was given primarily to avoid overwhelming participants with feedback, with a view to contacting these participants only if they had actionable findings. They had to consent to their general practitioner (GP) being informed about them.

### Neuroimaging

Participants underwent brain imaging on a single Biograph mMR 3 Tesla PET/MRI scanner (Siemens Healthcare). Participants were injected via an intravenous cannula with the $^{18}$F amyloid PET ligand florbetapir at the start of the imaging session, and dynamic amyloid data were obtained over 60 min. MRI data were acquired simultaneously, including volumetric T1-weighted, T2-weighted and fluid-attenuated inversion recovery (FLAIR) sequences; resting-state functional MRI; multishell diffusion-weighted imaging; three-dimensional gradient echo sequence for T2*-weighted/susceptibility weighted imaging; and arterial spin labelling (non-invasive perfusion imaging).

### Blood tests

Participants provided blood samples for standard clinical tests including haemoglobin, platelet count, vitamin $B_{12}$, urea, creatinine, random glucose and thyroid stimulating hormone (TSH). Samples were also taken for biomarker and genetic testing. Results of the clinical blood tests were reported back to the study team via email within 24 hours. Samples for biomarker and genetic testing were stored for future analysis.

### Duty of care protocol for neuroimaging

Given that Insight 46 participants were scanned at a single centre with availability of consultant neuroradiologists, and due to the unique nature of the cohort, it was decided that MRI scans would have a radiologist review. All T1-weighted, T2-weighted and FLAIR MRI sequences were reviewed by one of two consultant neuroradiologists within two weeks of the scan. Other sequences were not routinely reviewed on the basis that they do not form part of a standard diagnostic MRI examination in clinical practice. Neuroradiologists used a list of prespecified reportable and non-reportable

**Table 1** List of reportable and non-reportable MRI abnormalities (adapted from the UK Biobank, German National Cohort and Rotterdam Scan studies)[5 10 11]

| Reportable findings | Non-reportable findings |
|---|---|
| ► Acute brain infarction.<br>► Acute brain haemorrhage (note: not old bleeds).<br>► Intracranial mass lesions (note: not meningiomas in locations considered unlikely to cause problems).<br>► Suspected intracranial aneurysm or vascular malformation (including cavernomata) (note: not aneurysms <7 mm in diameter).<br>► Colloid cyst of the third ventricle.<br>► Acute hydrocephalus.<br>► Significant sinus disease with suspicion of underlying pathology (eg, unilateral sinus opacification).<br>► Other unexpected, serious or life-threatening findings. | ► White matter hyperintensities.<br>► Suspected demyelination.<br>► Non-acute brain infarction.<br>► Chronic hydrocephalus.<br>► Asymmetric ventricles.<br>► Lipoma of the corpus callosum.<br>► Developmental abnormalities.<br>► Enlarged perivascular spaces.<br>► Chiari malformation.<br>► Hippocampal or other focal atrophy. |

abnormalities to flag scans as being potentially reportable (table 1). This list was adapted from the UK Biobank study, which classified findings as reportable if they were potentially serious (i.e. life-threatening or likely to have a major impact on quality of life or function), based primarily on work performed by the German National Cohort.[5 10] Aneurysms <7 mm were not considered reportable in keeping with the Rotterdam Scan Study.[11] Neuroradiologists were also encouraged to flag scans with other unexpected findings if there was any possibility that further assessment might be required.

The reporting process was performed electronically using the web-based data management tool XNAT (www.xnat.org), thereby providing an audit trail (figure 1). Reporting radiologists downloaded images from the XNAT server, reviewed them and then completed a radiological read report within XNAT (online supplementary file 3). This took around ten minutes per scan. Radiologists were not given any clinical information regarding participants, other than knowing that they were all born in 1946. If a scan was flagged as potentially reportable, the study coordinator was automatically notified, and a multidisciplinary meeting was organised within four weeks of the study visit. The reporting neuroradiologist, study chief investigator and other relevant members from the study team were present at this meeting.

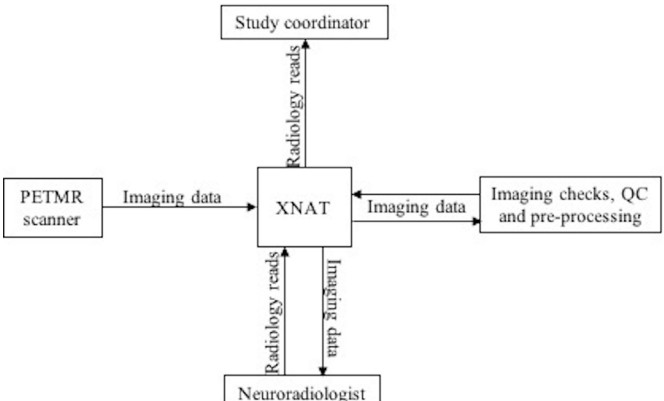

**Figure 1** Simplified overview of the process for viewing and reporting scans using XNAT. QC, quality control; PETMR, positron emission tomography and magnetic resonance

If the abnormality was agreed to meet the criteria for being reportable, the team decided on a clinical action plan (eg, further imaging and/or specialist referral). A study doctor then contacted the participant and their GP, by telephone and in writing, providing them with information about the MRI abnormality and the recommended clinical action. Since data were collected in an anonymised form, it was not possible to share the images for clinical use.

Results of the amyloid PET scan were not fed back to participants because of the diagnostic and prognostic uncertainties of using this test in cognitively normal individuals and lack of disease-modifying treatments for people with amyloid pathology. These ethical considerations have been discussed elsewhere.[12]

### Duty of care protocol for blood tests

Results of the clinical blood tests were reviewed by the study doctor and reported back to the participant's GP in writing within two weeks of the study visit. The participant was also sent a copy of these results if they had previously stated that they wished to receive one. If results fell outside the normal reference range (table 2), these abnormalities were highlighted in a letter sent to both the participant and their GP, and participants were advised to discuss them with their GP. If results were deemed to be significantly abnormal, falling beyond prespecified urgent action levels (table 2), the study doctor contacted the participant and their GP by telephone within 48 hours of the study visit. These prespecified levels were adapted from those used at the NSHD whole cohort sweep at age 60–64.[6] They reflect values at which urgent action would be warranted in clinical practice and were developed in consultation with clinical scientists and physicians in the relevant field. Biomarker and genetic test results were not reported back to participants.

### Follow-up of incidental findings on brain MRI

While participants have not been systematically followed up with regard to findings detected on brain MRI, data regarding outcomes have been obtained via different sources, mainly through telephone or written communication from participants, or through letters obtained from healthcare professionals.

## Table 2  Clinical blood tests, their normal reference ranges and urgent action levels

| Blood test | Normal reference range | Urgent action level |
|---|---|---|
| Haemoglobin (male), g/L | 130–170 | <100 or >200 |
| Haemoglobin (female), g/L | 115–155 | <100 or >200 |
| Platelets, ×10$^9$/L | 150–400 | <100 or >1000 |
| Vitamin B$_{12}$, pg/mL | 191–900 | <100 |
| Urea, mmol/L | 1.7–8.3 | >20 |
| Creatinine (male), µmol/L | 66–112 | >200 |
| Creatinine (female), µmol/L | 49–92 | >200 |
| Glucose, mmol/L | 3.5–10 | >20 |
| Thyroid stimulating hormone, mIU/L | 0.27–5.5 | <0.1 or >10 |

### Analysis

Data regarding the number and types of incidental findings, and the actions taken by the study team in response to them, were summarised as counts and percentages, and 95% CIs for proportions were calculated using the exact Clopper-Pearson method. Sex differences were assessed using a two-tailed two-sample test of proportions. A p value <0.05 was considered significant. For brain MRI analyses, participants without a scan were excluded. For blood result analyses, participants were excluded if they had a missing value for the specific test or category being analysed. Very few participants had missing blood result values, primarily due to sampling or processing errors, and these were assumed to have occurred at random. All analyses were performed in STATA V.14.2. Estimated glomerular filtration rate (eGFR) was derived using the Modification of Diet in Renal Disease study equation: GFR (mL/ min/1.73 m$^2$)=175× (Scr/88.4)$^{-1.154}$× (age)$^{-0.203}$ (× 0.742 if female), where Scr is serum creatinine in µmol/L.

### Participant involvement

Study members helped in the design of the Insight 46 study through participation in focus groups. Participants were invited to complete evaluation forms following their study visit, outlining any positive or negative aspects of their experience. Results from the Insight 46 study will be disseminated to participants through newsletters and public engagement events.

## RESULTS

502 participants attended a study visit in London from throughout mainland Britain between May 2015 and January 2018. The mean age was 70.7 (SD: 0.7) years and 49% were female. In total, 181 participants had a reportable incidental finding on either brain MRI or clinical blood tests, and 45 participants had more than one reportable finding.

### Brain MRI

93.8% of participants completed a brain scan (n=471). The most common reason for non-completion was claustrophobia (n=25). Other reasons included: being unable to lie comfortably in the scanner (n=3); concerns about radiation (n=1); possible metallic implants (n=1); and withdrawal from the study (n=1). 7.6% of scans (n=36) were flagged by neuroradiologists as having potentially reportable abnormalities for review. Following discussion between the reporting neuroradiologist and study chief investigator, 58.3% of these scans (n=21) were deemed to have an abnormality that fulfilled the criteria for being reportable. Therefore, in total, 4.5% of all scans had an incidental finding that was reported to the participant and their GP. Details of flagged findings that were not deemed reportable are listed in online supplementary file 4.

## Table 3  Number and percentage of reportable MRI abnormalities by type and sex

| | All (N=471) | | Male (N=241) | | Female (N=230) | |
|---|---|---|---|---|---|---|
| | n | % (95% CI) | n | % (95% CI) | n | % (95% CI) |
| Any abnormality | 21 | 4.5 (2.8 to 6.7) | 6 | 2.5 (0.9 to 5.3) | 15 | 6.5 (3.7 to 10.5) |
| Acute brain infarction | – | – | – | – | – | – |
| Acute brain haemorrhage | – | – | – | – | – | – |
| Suspected intracranial mass lesion | 7 | 1.5 (0.6 to 3.0) | 2 | 0.8 (0.1 to 3.0) | 5 | 2.2 (0.7 to 5.0) |
| Suspected intracranial aneurysm or vascular malformation | 9 | 1.9 (0.9 to 3.6) | 2 | 0.8 (0.1 to 3.0) | 7 | 3.0 (1.2 to 6.2) |
| Colloid cyst of the third ventricle | – | – | – | – | – | – |
| Acute hydrocephalus | – | – | – | – | – | – |
| Significant sinus pathology | 3 | 0.6 (0.1 to 1.9) | 1 | 0.4 (0.0 to 2.3) | 2 | 0.9 (0.1 to 3.1) |
| Other* | 2 | 0.4 (0.0 to 1.5) | 1 | 0.4 (0.0 to 2.3) | 1 | 0.4 (0.0 to 2.4) |

*Possible keratocystic odontogenic tumour of the right mandible (n=1); hyperintense area in the suprasellar cistern with differential diagnosis of small dermoid cyst, craniopharyngioma or thrombosed anterior communicating artery aneurysm (n=1).

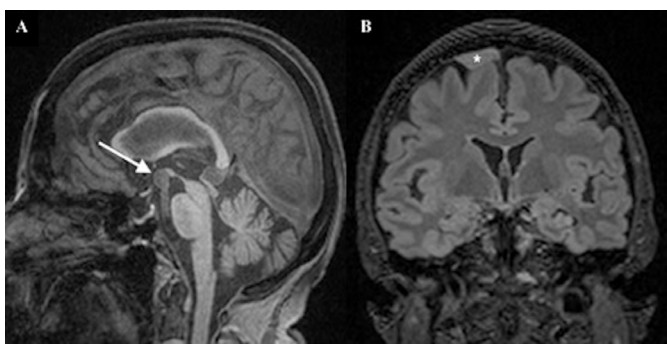

**Figure 2** (A) Sagittal T1-weighted image demonstrating a 10 mm aneurysm (arrow) arising from the tip of the basilar artery. (B) Coronal FLAIR image demonstrating a broad-based, extra-axial lesion (asterisk) overlying the right superior frontal gyrus, consistent with a meningioma. FLAIR, fluid-attenuated inversion recovery.

Table 3 summarises the number and percentage of reportable MRI abnormalities by type and sex. The most common abnormalities were suspected vascular malformations and suspected intracranial mass lesions, which were detected in 1.9% (n=9) and 1.5% (n=7) of participants respectively. Suspected cerebral aneurysms were the most common vascular abnormality, affecting 1.1% of participants (n=5; figure 2A). Suspected meningiomas were the most common intracranial lesion, affecting 0.6% of participants (n=3; figure 2B). Women were more likely to have a reportable MRI abnormality than men (6.5% vs 2.5%; p=0.034).

With regard to management of incidental findings, further imaging was recommended in 66.6% of cases (n=14); specialist referral was advised in 57.1% of cases (n=12); advice regarding medication and management was given in 19% of cases (n=4); and no action was recommended in 9.5% of cases where the abnormalities were found to be pre-existing and already being managed by the participant's local health services (n=2). Further information regarding follow-up and subsequent outcomes is summarised in online supplementary file 5.

### Standard clinical blood tests

Venepuncture was successful in over 99% of participants (n=498). Almost all participants chose to receive a copy of their clinical blood test results (n=496). There were missing blood result values in some participants (n=9) due to insufficient samples, lab errors, clumped platelets or a clotted sample. Of participants with complete blood result data, 34.6% had at least one abnormality on standard clinical blood tests (n=169). Of those participants with abnormalities, urgent action was required for 6.5% (n=11). In many of these cases (n=6), the participant's GP confirmed that the abnormality was pre-existing and already being managed. Table 4 summarises the number and percentage of blood test abnormalities by type and sex. Overall, men were significantly more likely to have at least one blood test abnormality than women (40.8% vs 28.0%; p=0.003). However, removing 'low creatinine' as an abnormality resulted in there being no significant difference between men and women (30.8% vs 26.4%; p=0.277).

**Table 4** Number and percentage of clinical blood test abnormalities by type and sex

| | All (N=498) | | Male (N=255) | | Female (N=243) | |
|---|---|---|---|---|---|---|
| | n/N | % (95% CI) | n/N | % (95% CI) | n/N | % (95% CI) |
| Any abnormality | 169/489 | 34.6 (30.3 to 39.0) | 102/250 | 40.8 (34.6 to 47.2) | 67/239 | 28.0 (22.4 to 34.2) |
| Polycythaemia | 15/494 | 3.0 (1.7 to 5.0) | 11/254 | 4.3 (2.2 to 7.6) | 4/240 | 1.7 (0.5 to 4.2) |
| Anaemia | 19/494 | 3.8 (2.3 to 5.9) | 14/254 | 5.5 (3.0 to 9.1) | 5/240 | 2.1 (0.7 to 4.8) |
| Thrombocytosis | 10/492 | 2.0 (1.0 to 3.7) | 2/252 | 0.8 (0.1 to 2.8) | 8/240 | 3.3 (1.4 to 6.5) |
| Thrombocytopaenia | 11/492 | 2.2 (1.1 to 4.0) | 9/252 | 3.6 (1.6 to 6.7) | 2/240 | 0.8 (0.1 to 3.0) |
| Elevated vitamin $B_{12}$ | 10/495 | 2.0 (1.0 to 3.7) | 5/253 | 2.0 (0.6 to 4.6) | 5/242 | 2.1 (0.7 to 4.8) |
| Low vitamin $B_{12}$ | 16/495 | 3.2 (1.9 to 5.2) | 6/253 | 2.4 (0.9 to 5.1) | 10/242 | 4.1 (2.0 to 7.5) |
| Elevated urea | 40/497 | 8.0 (5.8 to 10.8) | 23/254 | 9.1 (5.8 to 13.3) | 17/243 | 7.0 (4.1 to 11.0) |
| Elevated creatinine | 17/497 | 3.4 (2.0 to 5.4) | 10/254 | 3.9 (1.9 to 7.1) | 7/243 | 2.9 (1.2 to 5.8) |
| Low creatinine | 41/497 | 8.2 (6.0 to 11.0) | 33/254 | 13.0 (9.1 to 17.7) | 8/243 | 3.3 (1.4 to 6.4) |
| eGFR <60* | 43/497 | 8.7 (6.3 to 11.5) | 15/254 | 5.9 (3.3 to 9.6) | 28/243 | 11.5 (7.8 to 16.2) |
| Hyperglycaemia | 21/497 | 4.2 (2.6 to 6.4) | 16/254 | 6.3 (3.6 to 10.0) | 5/243 | 2.1 (0.7 to 4.7) |
| Hypoglycaemia | 5/497 | 1.0 (0.3 to 2.3) | 1/254 | 0.4 (0.0 to 2.2) | 4/243 | 1.6 (0.5 to 4.2) |
| Elevated TSH | 13/496 | 2.6 (1.4 to 4.4) | 4/253 | 1.6 (0.4 to 4.0) | 9/243 | 3.7 (1.7 to 6.9) |
| Low TSH | 9/496 | 1.8 (0.8 to 3.4) | – | – | 9/243 | 3.7 (1.7 to 6.9) |
| Urgent action | 11/489 | 2.2 (1.1 to 4.0) | 3/250 | 1.2 (0.2 to 3.5) | 8/239 | 3.3 (1.5 to 6.5) |

NB, Participants were excluded if they had a missing value for the specific test or category being analysed.
*eGFR (mL/min/1.73 m$^2$) was calculated to facilitate comparison with other studies; it was not reported back to participants.
eGFR, estimated glomerular filtration rate; TSH, thyroid stimulating hormone.

## DISCUSSION

In this study of older adults, aged 69–71, reportable incidental findings on brain MRI were present in 4.5% of scanned participants, with suspected vascular malformations and suspected intracranial mass lesions present in 1.9% and 1.5% of participants respectively. Clinical blood test abnormalities were common, affecting around one-third of participants. However, very few blood test abnormalities required urgent action, and many of those that did were previously known to the participants' GPs and had already been acted on.

### Comparison with other studies

Due to the recent proliferation of neuroimaging research, incidental findings on brain MRI are often reported in the literature.[8 9] The reported prevalence varies between studies, likely reflecting differences in the definition of what constitutes an incidental finding, as well as variability in participant demographics and imaging protocols. Many imaging studies do not require routine review of all scans by a radiologist, and researchers will only ask for a radiologist opinion if an abnormality is identified incidentally by a radiographer during scanning or by researchers during data analysis.[13 14] Such studies may have lower detection rates, but are presumably less likely to publish data on incidental finding prevalence.

A 2018 systematic review reported an overall prevalence of 1.4% (95% CI 1.0% to 2.1%) for potentially serious brain incidental findings.[8] This is somewhat lower than the 4.5% (95% CI 2.8% to 6.7%) detected in Insight 46 participants, although this review consisted mainly of studies with younger participants using MRI scanners of 1.5 Tesla or less. Most of the studies in this review used at least one radiological reader. Another systematic review reported a much higher prevalence of 22% (95% CI 14% to 31%), likely due to their inclusion of all findings, regardless of their clinical seriousness.[9] Comparing specific abnormalities, namely suspected intracranial mass lesions and vascular malformations, in Insight 46 and 1936 Lothian Birth Cohort (LBC) revealed similar rates: 1.4% (95% CI 0.7% to 2.6%) and 2% (95% CI 1.1% to 3.3%) respectively in LBC subjects aged 73, compared with 1.5% (95% CI 0.6% to 3.0%) and 1.9% (95% CI 0.9% to 3.6%) in Insight 46.[15] Results from another large population-based study, which included over 5800 subjects with a mean age 64.9 years, were marginally higher than Insight 46, with a prevalence of 2.5% (95% CI 2.1% to 2.9%) for suspected meningiomas and 2.3% (95% CI 2.0% to 2.7%) for suspected cerebral aneurysms.[11]

Most previous studies have found no significant difference in prevalence of potentially serious brain MRI findings by sex.[8] In Insight 46, however, higher rates were observed in female versus male participants. This was primarily driven by greater numbers of suspected intracranial mass lesions and vascular abnormalities in women, possibly due to the fact that meningiomas and cerebral aneurysms are more common in women than in men.[11]

With regard to blood tests, Insight 46 tended to have either similar or lower rates of abnormalities than other studies. The prevalence of anaemia in a systematic review of studies involving community-dwelling older adults was 12%, which is somewhat higher than the 3.8% (95% CI 2.3% to 5.9%) detected in Insight 46 participants.[16] The prevalence of chronic kidney disease stages 3–5 (eGFR $<60\,mL/min/1.73\,m^2$) is estimated to be around 6.1% in adults under 65 in England, rising to 13.5% for individuals aged 65–74, according to data collected in the 2009–2010 Health Survey for England and 2011 Census.[17] This is broadly in keeping with the rate of 8.7% (95% CI 6.3% to 11.5%) detected in Insight 46 participants. Vitamin $B_{12}$ deficiency was detected in around 5% of individuals aged 65–74 years old in a large UK-based study, compared with 3.2% (95% CI 1.9% to 5.2%) in Insight 46 participants.[18] Another large UK-based study found a prevalence of 7.9% (95% CI 6.4% to 9.6%) for elevated TSH and 6.0% (95% CI 4.7% to 7.4%) for low TSH in adults over 60 years old, somewhat higher than the 2.6% (95% CI 1.4% to 4.4%) and 1.8% (95% CI 0.8% to 3.4%) detected in Insight 46 participants.[19]

Discrepancies in the reported prevalence of blood test abnormalities between Insight 46 and other studies may be partly related to differences in laboratory assays, thresholds for defining abnormal values and participant demographics. However, it is also likely that certain blood test abnormalities are under-represented in Insight 46, since participants underwent clinical blood testing at a previous study visit aged 60–64 years old, and any abnormalities detected then were likely addressed at that time.[20] Indeed, comparing participant results at age 60–64 with those in the Insight 46 study revealed that only 2 out of 9 participants still had anaemia, 8 out of 27 still had an elevated TSH, and 5 out of 10 still had a low TSH.

### Strengths and weaknesses

A major strength of the Insight 46 study is that it involved a large number of participants who underwent brain imaging and blood testing, at an almost identical age, and received feedback regarding incidental findings according to a prespecified standardised protocol. These participants were all recruited from the NSHD, the longest running British birth cohort, which has remained broadly representative of the population born in mainland Britain in 1946.[21] Distance from London was not found to be predictive of participation.[22]

High-resolution MRI sequences were obtained using the same 3 Tesla PET/MRI scanner for all participants, and images were systematically reviewed by one of two experienced consultant neuroradiologists. The process of reviewing scans was user-friendly and automated where possible, allowing scans to be reported within a short timeframe, thereby reducing the workload of the neuroradiologists. Scans were sometimes flagged for review, despite not having a reportable finding according to the study protocol, usually because the radiologist felt that the abnormality was serious enough to warrant further discussion. This was encouraged in order to avoid overlooking findings that might be considered actionable in the appropriate clinical

context. In practice, however, this did not alter the number of findings reported to participants.

The duty of care protocol was developed in accordance with the MRC and Wellcome Trust framework on management of health-related research findings.[2] Any potentially serious brain MRI findings or blood test abnormalities were reported back to participants and their GPs, in keeping with the ethical principle of beneficence. Findings were not disclosed if tests lacked clinical utility or were not actionable, in order to minimise participant distress and harm. Participants were fully informed of the protocol for managing incidental findings as part of the consent process and were given the choice on whether they wanted to receive a copy of their blood results, thereby respecting their autonomy to make decisions about their own health. While it can be argued that research is generally not meant to benefit participants directly, many participants view medical input as an incentive to take part and there is an expectation that they will be informed of any serious findings. This needs to be balanced against the potential negative consequences of reporting incidental findings.[5]

A limitation of this study is that participant perception regarding the disclosure of incidental findings was not formally assessed, nor was the impact on their longer term health and psychological well-being. Many participants, however, gave informal feedback on post-visit evaluation forms that they appreciated being told about findings pertinent to their health and saw this as a benefit of being involved in the study. Moreover, almost all participants chose to receive a copy of their blood test results. These observations are consistent with the results of a study commissioned by the Wellcome Trust and MRC, which found overwhelming public support for the disclosure of incidental findings in research, particularly in relation to serious and treatable conditions.[23] This is also supported by the work of several other studies.[3 5 24 25]

A further limitation of Insight 46 is that NSHD participants are all white Caucasian and, due to changing population demographics, results may not be directly generalisable to the current British population aged 70, or indeed younger populations. Furthermore, in separate analyses of recruitment to Insight 46, NSHD participants with higher educational attainment, non-manual socioeconomic position and better self-rated health were more likely to take part.[22]

## Implications and future work

The findings of this study will be relevant to future studies involving older adults, including clinical trials of secondary prevention drugs for Alzheimer's disease, which often involve MRI-based outcome measures and blood monitoring. Awareness of the expected prevalence of incidental findings on brain MRI and clinical blood tests in this age group, based on predefined protocols for their determination, should allow researchers to be better prepared for managing them and participants to be better informed of their likelihood as part of the consent process.

The findings also have implications for clinical practice. In patients with benign-sounding headaches and normal neurological examination, for example, the chances of finding a serious intracranial cause on brain imaging is less than 1%.[26 27] Nonetheless, patients presenting with chronic headache frequently undergo brain imaging, usually to provide reassurance, and often at the patient's own request. These patients are rarely consented for the risk of discovering an incidental finding, despite the potential negative consequences. Greater awareness of the expected frequency and nature of incidental findings on brain imaging and blood tests should allow clinicians to counsel patients regarding their probability, and to balance this risk against the potential benefits of undergoing a test when deciding whether it is appropriate.

While the focus of this study was on potentially serious brain imaging findings, awareness of the prevalence of other incidental abnormalities, such as white matter disease, would also be useful from a clinical perspective. In separate analyses, the distribution of white matter disease burden in Insight 46 participants was found to be highly non-linear, making it difficult to define a threshold of abnormality.[28] Ongoing work investigating these changes, including longitudinal follow-up to assess their consequences, should help inform clinicians regarding their significance and management.

Further work is also needed to assess the implications of disclosing incidental findings in research studies, including the psychological effects and longer term clinical consequences, as well as the impact on research integrity, particularly in longitudinal population studies where disclosure might lead to a biased sample. Outcome data regarding incidental brain MRI findings in Insight 46 were obtained, but this does not represent final diagnoses for all participants, nor does it include assessment of emotional impact. Due to the longitudinal nature of this study, it will be possible to collect these outcomes in a more systematic way after a longer interval. This will be helpful to inform debates on the ethics of feeding back incidental findings to participants, adding to the work of several other ongoing studies.[3–5 11 24 25]

## Author affiliations

[1]Dementia Research Centre, UCL Queen Square Institute of Neurology, London, UK
[2]Lysholm Department of Neuroradiology, The National Hospital for Neurology and Neurosurgery, London, UK
[3]MRC Unit for Lifelong Health and Ageing, University College London, London, UK
[4]School of Biomedical Engineering and Imaging Sciences, King's College London, London, UK
[5]Leonard Wolfson Experimental Neurology Centre, UCL Queen Square Institute of Neurology, London, UK
[6]Department of Brain Repair and Neurorehabilitation, UCL Queen Square Institute of Neurology, London, UK
[7]Institute of Nuclear Medicine, University College London Hospitals, London, UK

**Acknowledgements** We are very grateful to those study members who helped in the design of the study through focus groups, and to the participants both for their contributions to Insight 46 and for their commitment to research over the last seven decades. We are grateful to the radiographers and nuclear medicine physicians (Professor Ashley Groves, Dr Jamshed Bomanji, Dr Irfan Kayani) at the UCL Institute of Nuclear Medicine, and to the staff at the Leonard Wolfson Experimental Neurology Centre at UCL. We would like to acknowledge Dan Marcus and Rick Herrick for assistance with XNAT, Dr Philip Curran for assistance with data sharing

with the MRC Unit for Lifelong Health and Ageing, the DRC trials team for assistance with imaging QC, Mark White for his work on data connectivity, and Suzie Barker for her assistance with research governance. Avid Radiopharmaceuticals (a wholly owned subsidiary of Eli Lilly) provided the PET amyloid tracer (florbetapir) but had no part in the design of the study.

**Contributors** SEK and JMS conceived the manuscript. TDP, CAL, AK, SEK, SMB, HM-S and AW recruited participants to the study. CH and SS reviewed and reported the MRI brain scans. TDP, CAL, AK, SEK, SMB, S-NJ, KL and JC contributed to data collection. DMC, IBM, DLT and AB were responsible for setting up the imaging acquisition protocols, image processing and quality control. DGB was involved in data management. SEK analysed the data and drafted the initial manuscript. JMS, NF and MR are co-principal investigators of the study. All authors critically revised the manuscript and approved the submitted version.

**Funding** Insight 46 is funded by grants from Alzheimer's Research UK (ARUK-PG2014–1946, ARUK-PG2017-1946; PIs JMS, NF and MR), the Medical Research Council Dementias Platform UK (CSUB19166; PIs JMS, NF and MR), The Wolfson Foundation (PR/ylr/18575; PIs NF and JMS), the Medical Research Council (MC_UU_12019/1, PI Kuh; MC_UU_12019/3, PI MR), the Wellcome Trust (Clinical Research Fellowship 200,109/Z/15/Z; TDP) and Brain Research Trust (UCC14191; PI JMS).

**Competing interests** NF's research group has received payment for consultancy or for conducting studies from Avid Radiopharmaceuticals, Biogen, Eisai, Elan, Eli Lilly Research Laboratories, GE Healthcare, IXICO, Janssen, Johnson & Johnson, Lundbeck, Pfizer, Roche, Sanofi-Aventis and Wyeth Pharmaceuticals. NF receives no personal compensation for the activities mentioned above. JMS has received research funding from Avid Radiopharmaceuticals (a wholly owned subsidiary of Eli Lilly), has consulted for Roche Pharmaceuticals, Biogen and Eli Lilly, has given educational lectures sponsored by GE, Eli Lilly and Biogen, and serves on a Data Safety Monitoring Committee for Axon Neuroscience SE.

**Patient consent for publication** Not required.

**Ethics approval** Ethical approval was granted by the National Research Ethics Service (NRES) Committee London (REC reference 14/LO/1173; PI JMS).

**Provenance and peer review** Not commissioned; externally peer reviewed.

**Data sharing statement** Data are available upon reasonable request.

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
