## [Reviewer comments · BMJ Open]

ARTICLE DETAILS

TITLE (PROVISIONAL)	Incidental findings on brain imaging and blood tests: results from the first phase of Insight 46, a prospective observational sub-study of the 1946 British birth cohort
AUTHORS	Keuss, Sarah; Parker, Thomas; Lane, Christopher; Hoskote, Chandrashekar; Shah, Sachit; Cash, David; Keshavan, Ashvini; Buchanan, Sarah; Murray-Smith, Heidi; Wong, Andrew; James, Sarah-Naomi; Lu, Kirsty; Collins, Jessica; Beasley, Daniel; Malone, Ian; Thomas, David; Barnes, Anna; Richards, M; Fox, Nick; Schott, Jonathan M.

VERSION 1 – REVIEW

REVIEWER	AJ Larner Walton Centre for Neurology and Neurosurgery, Liverpool, United Kingdom
REVIEW RETURNED	09-Apr-2019

GENERAL COMMENTS	Information on the prevalence of incidental findings on brain imaging and blood tests in older people is very welcome, not least because it is highly likely that increasing numbers will undergo these investigations as the population ages. Although this is a research study, with clearly defined protocols, the results have potential application to inform clinical practice. Chief among these may be pre-test counselling of patients on the risks of possible incidental test findings, a particular issue with brain imaging which may be requested by clinicians with little or no experience of image interpretation. Understandably the study has focussed on potentially serious imaging findings, but in clinical practice the finding of incidental white matter hyperintensities (a “non-reportable” finding here) often generates anxiety, sometimes being interpreted by non-neurological clinicians as indicative of “stroke” or even “vascular dementia”. It would have been useful from a clinical standpoint to have included data on the prevalence of this incidental finding. I would anticipate that such changes were seen in a very high percentage of the Insight 46 cohort. Potential shortcomings of the study relate to the generalizability of the findings. This was a research population selected by date of birth, one week in 1946, and location of birth, mainland Britain. No information on patient ethnicity is given but it might be anticipated that the cohort would include few participants of either Afro-Caribbean (pre-Windrush) or Asian ethnicity. The findings might not,
---

	therefore, be entirely representative of the current British population aged 70. The necessity of travel to London for the study investigations might mean that, although participants were drawn from throughout mainland Britain, those from England might be over-represented compared to those from Scotland and Wales (no details of patients' geographical origin are given). Minor points for correction: p10, line 56: "data was" to read "data were". Reference 17: "Grimely EJ" to read "Grimley Evans J". References 21 and 22: only 2 authors given before et al., unlike previous references.
--	---

REVIEWER	Lorna M. Gibson Usher Institute of Population Health Sciences and Informatics, University of Edinburgh, UK I received funding from the Wellcome Trust to study the epidemiology and impact of potentially serious incidental findings in the UK Biobank Imaging Study. I receive personal fees from UK Biobank for imaging consultancy work and am a member of the UK Biobank Imaging Working Group.
REVIEW RETURNED	05-May-2019

GENERAL COMMENTS	Thank you for the opportunity to review this interesting manuscript. Keuss et al. present the incidental findings detected on brain magnetic resonance imaging (MRI) and blood tests during the first phase of Insight 46, a longitudinal prospective sub-study of the 1946 British Birth Cohort. Empirical data on the frequency and types of incidental findings detected in healthy volunteers are extremely useful to practically inform future incidental findings feedback policies and the design of participant consent materials. My recent work has involved a systematic review of the prevalence and types of incidental findings on brain (as well as body) MRI, and a study of the impact of feedback of potentially serious incidental findings in the UK Biobank study, and I remain a member of UK Biobank's incidental findings team. This perspective clearly informs the comments I have made below, and which I hope are of use to the authors. Their work is interesting and relevant, but could be clarified in places and make much more use of the experience and insight of other pertinent studies. I have numbered my comments for ease of response.  1. The first bullet point of the strengths and limitations paragraph states that 'A large number of participants underwent blood testing and brain imaging, at an almost identical age, according to a pre-specified standardised protocol,' but research studies are usually conducted to a pre-specified standardised protocol. Do the authors mean to instead say that 'A large number of participants underwent blood testing and brain imaging, at an almost identical age, and received feedback of potentially serious incidental findings according to a pre-specified standardised protocol?' 2. On page 6/32, the statement 'There are, however, important ethical reasons for disclosing certain incidental findings to participants in appropriate circumstances, particularly when they relate to serious and potentially treatable conditions' could be enhanced by a further reference to Wolf et al. 2008 (listed as reference number 1).
--

	3. On page 6/32, the statement ‘Incidental findings often lead to anxiety and have the potential to lead to unnecessary and invasive procedures for study participants’ requires referencing as these outcomes have been demonstrated by teams from the UK Biobank (for transparency, I am a co-author of this study), the German National Cohort and the Netherlands Epidemiology of Obesity Study (de Boer et al., 2018; Gibson et al., 2018a; Schmidt et al., 2013). 4. On page 7/32, the statement ‘Several studies have reported rates of incidental findings in different samples previously,[5-10]’ could be brought up to date with references to two recent systematic reviews (again, for transparency, I am a co-author of one of these studies) (Gibson et al., 2018b; O’Sullivan et al., 2018), rather than selected studies included in those two reviews. 5. On page 8/32 within the ‘recruitment’ subsection, can the authors provide more detail on what constituted ‘relevant life course data’ to allow readers to judge whether or not this selection criterion may have introduced bias. 6. On page 8/32, the ‘consent’ subsection could be expanded to provide some important detail which would inform current debates on ethical feedback policies. In particular, please could the authors provide: a link, reference or supplementary file containing the study consent form and a short description or quote in this section on the wording of the consent form with regard to the participants’ consent to receiving feedback of incidental findings; the wording for the option to ‘opt out’ of receiving correspondence about blood test results (and presumably the lack of option to ‘opt out’ of receiving correspondence about brain MRI incidental findings?); justification for the offering of this ‘opt-out’ option; details on whether or not participants were made aware that a lack of feedback did not constitute a health check, given the limitations of the research imaging for providing firm clinical diagnoses. 7. On page 8/32, the final line should read ‘MRI data were acquired simultaneously.’ 8. On page 9/32, the first use of ‘FLAIR’ needs to be defined in full. 9. On page 9/32, the ‘duty of care protocol for neuroimaging’ section could also be expanded. Please could the authors provide details on (or a reference to) how they arrived at the justification for consultant neuroradiologist review of all images for their study, as later on page 18 they state that ‘Most imaging studies, for example, do not require routine review of all scans by a radiologist.[13]’ 10. The list of reportable and non-reportable findings has been adapted from the UK Biobank study lists (which were originally adapted from work by the German National Cohort),(Bertheau et al., 2016) but the size cut-off for feedback of intracranial aneurysms appears to have come from the Rotterdam Scan Study protocols (> 7 mm) (Bos et al., 2016). These three studies should be referenced in the text and in a footnote to Table 1. 11. On page 10/32, I would like to see more detail on the radiologists’ reporting processes. Specifically, did the ‘radiological read report’ consist of the findings listed in Table 1 being marked as
--	--

present or absent (i.e. reporting to a pre-specified checklist)? In addition, please provide details on the clinical information provided to the radiologists to aid the interpretation of images (e.g. age, sex, clinical history, previous imaging), as this will likely influence the prevalence of incidental findings.

12. On page 10/32, the second to last line should read 'and then recommended clinical action.'

13. On page 10/32, the authors state that 'since data was [sic] collected in an anonymised form, it was not possible to share the images for clinical use.' This is in contrast to UK Biobank (GPs can request copies of images for participants with potentially serious incidental findings), and could the authors please explain this further.

14. On page 11/32, could the authors clarify whether blood test results outside the normal reference range were fed back only to participants, or to participants and their GPs?

15. On page 12/32, before the analysis section, can the authors insert a sentence to state whether or not participants were systematically followed up, as the results section is inconsistent in reporting findings as 'suspected intracranial mass lesions' versus 'intracranial mass lesions,' and it is not clear if the latter description reflects the initial incidental finding appearance, or a final diagnosis. During our systematic review of studies of potentially serious findings on brain (and body) MRI, we found limited reporting of firm final diagnoses generated after systematic follow-up, and during our study of UK Biobank participants, around 80% of findings detected by radiologists turned out not to be serious, i.e. many descriptions of incidental findings changed substantially after follow-up. Throughout this current manuscript it is important to clearly distinguish between descriptions of the incidental finding (i.e. a suspected condition demonstrated on non-diagnostic imaging), versus a final diagnosis of disease; this would set this work apart from other studies in terms of clarity of reporting.

16. On page 12/32, the authors need to add: their methods for calculating 95% confidence intervals (e.g. an exact method which assumes a binomial distribution, given the data are of prevalence); whether they conducted one- or two-tailed significance tests; the p-value they considered significant; how they dealt with missing data (e.g. the reduced denominators detailed in Table 4, and the participants who did not complete imaging (6.2%)).

17. On page 12/32, the authors describe the participants' ages as '70.7 (+/-0.7).' Is this the standard deviation? Please clarify.

18. Can the authors provide details (in a supplement perhaps) of the 15 findings initially flagged by the neuroradiologists and deemed not for feedback (i.e. 42% of the findings flagged by the consultants) and add a sentence to the discussion to describe why these did not (or why the 58% did) make it through the committee decision. Do the authors have any reflections on the study's incidental findings process as a result of this, given the large proportion of consultant-flagged findings which were not eventually fed back?

19. On page 13/32, please state how many participants received feedback of a reportable finding from either brain MRI or

blood tests, in order to clearly describe how many people were affected by feedback (of any type) during Insight 46. Also please specify how many participants had more than one reportable finding (from either brain, blood tests, or across both tests), again to illustrate the burden on participants.

20. Our systematic review did not find any convincing evidence of a difference in prevalence of potentially serious incidental findings on brain and body MRI between men and women (Gibson et al., 2018b). Please add one or two sentences in your discussion about your findings of differences in prevalence of reportable incidental findings on both the brain MRI and the blood tests between men and women.

21. On page 13/32, please clarify if the most common vascular abnormalities were cerebral aneurysms, or suspected (or likely?) cerebral aneurysms, and if the most common intracranial lesions were meningioma or suspected meningioma, to clearly distinguish between a description of an incidental finding and of a final diagnosis.

22. It is not clear whether or not (selected or unselected groups of) participants were followed up after they received feedback, how this was done, and over what time period. Please can you clarify in this in your methods section, as the final paragraph on page 13/32 (which extends on to page 14/32) and in the section on results of blood tests, you provide considerable detail on the outcomes of what appears to be a selected group of participants. We have recently documented that few studies seem to systematically follow up participants who receive feedback of incidental findings (Gibson et al., 2018b), and if the Insight 46 study did indeed conduct this, the methods and results should be much clearer as this would be a valuable contribution to the field. If however, only selected groups of participants were followed up, and the text reflects some of the more interesting cases, please be clear about this and state in the discussion that the data do not accurately reflect the burden of follow-up experienced by the entire group, nor do they allow for quantification of the numbers and types of follow-up and that more data are clearly needed on this to inform judgements on the benefits and harms of feedback of incidental findings.

23. Your finding that few participants opt out of feedback of incidental findings on blood tests is similar to the finding from the German National Cohort (Hegenscheid et al., 2013) that hardly any participants opted out of feedback of imaging findings, and a reference should be included in the 'comparison with other studies' section of the discussion.

24. Please add a short footnote to table 4 to explain the variability in denominators and how the missing data were dealt with so that this table can stand alone independently from the text.

25. The headings called 'Number' in Table 4 would be more accurately labelled 'n/N.'

26. I do feel that the 'Comparison with other studies' section should begin with references to the two recent large systematic reviews of prevalence of incidental findings on brain MRI I mentioned above (Gibson et al., 2018b; O'Sullivan et al., 2018) rather than a selection of five studies contained within these. You are correct in stating that the prevalence reported in our review is

lower than in the current Insight 46 cohort, although O'Sullivan et al. found far higher prevalences and you may wish to discuss your work in light of their result too.

27. You state that 'Most imaging studies, for example, do not require routine review of all scans by a radiologist.[13] Often, researchers will only ask for a radiologist opinion if an abnormality is identified incidentally by a radiographer during scanning or by researchers during data analysis.' In contrast to your statement, the vast majority of studies included in our review used at least one radiologist reader, and you do not provide any references to support your second statement. While the UK Biobank does employ a process of radiographer flagging of concerning images for a radiologist to review (Gibson et al., 2018a), the German National Cohort train radiologists to read all their imaging (Bertheau et al., 2016), and the Rotterdam Scan Study train readers to review all images of which those scans with concerning findings are then reviewed by two neuroradiologists. I think your two sentences quoted above are worth clarifying and referencing, as the protocols for incidental findings vary considerably (as is appropriate), depending on the context of the study.

28. The prevalence estimates of incidental findings detected during the Rotterdam Scan Study (reference number 6), which you describe on page 19/32, have been updated (Bos et al., 2016) and your text should reflect this.

29. In the 'strengths and weaknesses' subsection on page 20/32, I think again the authors mean to say 'A large number of participants underwent blood testing and brain imaging, at an almost identical age, and received feedback of potentially serious incidental findings according to a pre-specified standardised protocol.'

30. On page 19/32, please change 'images were systematically reviewed by two experienced consultant neuroradiologists' to 'images were systematically reviewed by one of two experienced consultant neuroradiologists' so it is consistent with your methods section.

31. On page 19/32 you describe your feedback process as 'in keeping with the ethical principle of beneficence.' It can be argued that participants in research studies do not stand to benefit from participating, nor is there convincing evidence of benefit (although there is evidence of real harm) of feeding back incidental findings (Gibson et al., 2018a). Indeed, the research imaging performed in your study is not a 'health check.' Please expand on your reasoning here.

32. On page 21/32 you mention that 'participant perception regarding the disclosure of incidental findings was not formally assessed, nor was the impact on their longer-term health and psychological wellbeing.' Please add text here to say that data on final diagnoses were not systematically collected (if this is the case). Also, if there are plans to assess these outcomes in the Insight 46 cohort please do describe this here, as this would be very useful to inform debates of the benefits and harms of feedback of incidental findings.

33. Your statement on page 21/32 ('Many participants, however, gave informal feedback on post-visit evaluation forms that they

appreciated being told about findings pertinent to their health and saw this as a benefit of being involved in the study') is consistent with the findings of the UK Biobank, the German National Cohort, and the Netherlands Epidemiology of Obesity Study and these would be worth referencing (de Boer et al., 2018; Gibson et al., 2018a; Hegedus et al., 2019).

34. On page 21/32, you state 'Moreover, almost all participants chose to receive a copy of their blood test results. These observations are consistent with results of a study commissioned by the Wellcome Trust and MRC, which found overwhelming public support for the disclosure of incidental findings in research, particularly in relation to serious and treatable conditions.[20]' But you state in your methods 'Results of the clinical blood tests were reviewed by the study doctor and reported back to the participant and their GP in writing within two weeks of the study visit.' My reading of this is that everyone received feedback of all blood test results. If that is the case, then I wonder instead if your findings reflect participants' wishes to receive any feedback, regardless of seriousness (which would be need references to different studies which demonstrate this (Kirschen et al., 2006)). Please clarify.

35. In the 'implications and future work' section, you state that 'By defining what is actionable and providing the expected prevalence of incidental findings on brain MRI and clinical blood tests in this age group, researchers may be better prepared for managing incidental findings, and participants [sic] better informed of their likelihood as part of the consent process.' However the lists of findings used by Insight 46 are based extremely closely on the work of other studies (initially an extensive piece of work by the German National Cohort, which was then adapted by UK Biobank, which I think you have further adapted and combined with the Rotterdam Scan Study policy). In light of this I do not think you can claim that Insight 46 has defined what is actionable in terms of a finding for feedback. Instead, you can certainly say that Insight 46 has demonstrated that the work of these other studies (predominantly the German National Cohort, to give credit where it is due) appears to be adaptable and feasible to use in other imaging contexts.

36. Your closing statement describes work which continues to be conducted by several large European studies I have already mentioned above (among others), some of which is already published (Bos et al., 2016; de Boer et al., 2018; Gibson et al., 2018a; Hegedus et al., 2019; Hegenscheid et al., 2013; Schmidt et al., 2013), and you should reference them. A stronger closing statement may describe how your current manuscript also feeds in to the debate on ethical feedback policies and design of the consent materials (specifically those used in the Insight 46 study, or perhaps birth cohorts more generally).

References

Bertheau RC, von Stackelberg O, Weckbach S, Kauczor H-U, Schlett CL. 2016. Management of incidental findings in the German National Cohort. In: Weckbach S, editor. *Incidental radiological findings*. First ed. Cham, Switzerland: Springer.

Bos D, Poels MM, Adams HH, Akoudad S, Cremers LG, Zonneveld HI, Hoogendam YY, Verhaaren BF, Verlinden VJ, Verbruggen JG,

	Peymani A, Hofman A, Krestin GP, Vincent AJ, Feelders RA, Koudstaal PJ, van der Lugt A, Ikram MA, Vernooij MW. 2016. Prevalence, clinical management, and natural course of incidental findings on brain MR images: the population-based Rotterdam Scan Study. Radiology 281(2):507-515. de Boer AW, Drewes YM, de Mutsert R, Numans ME, den Heijer M, Dekkers OM, de Roos A, Lamb HJ, Blom JW, Reis R. 2018. Incidental findings in research: a focus group study about the perspective of the research participant. J Magn Reson Imaging 47(1):230-237. Gibson LM, Littlejohns TJ, Adamska L, Garratt S, Doherty N, UK Biobank Imaging Working Group, Wardlaw JM, Maskell G, Parker M, R. B, Matthews PM, Collins R, Allen NE, Sellors J, Sudlow CLM. 2018a. Impact of detecting potentially serious incidental findings during multi-modal imaging [version 3; referees: 2 approved, 1 approved with reservations]. Wellcome Open Res 2:114. Gibson LM, Paul L, Chappell FM, Macleod M, Whiteley WN, Salman RA, Wardlaw JM, Sudlow CLM. 2018b. Potentially serious incidental findings on brain and body magnetic resonance imaging of apparently asymptomatic adults: systematic review and meta-analysis. BMJ 363:k4577. Hegedus P, von Stackelberg O, Neumann C, Selder S, Werner N, Erdmann P, Granitza A, Volzke H, Bamberg F, Kaaks R, Bertheau RC, Kauczor HU, Schlett CL, Weckbach S. 2019. How to report incidental findings from population whole-body MRI: view of participants of the German National Cohort. Eur Radiol. (Epub ahead of print) Hegenscheid K, Seipel R, Schmidt CO, Volzke H, Kuhn JP, Biffar R, Kroemer HK, Hosten N, Puls R. 2013. Potentially relevant incidental findings on research whole-body MRI in the general adult population: frequencies and management. Eur Radiol 23(3):816-826. Kirschen MP, Jaworska A, Illes J. 2006. Subjects' expectations in neuroimaging research. J Magn Reson Imaging 23(2):205-9. O'Sullivan JW, Muntinga T, Grigg S, Ioannidis JPA. 2018. Prevalence and outcomes of incidental imaging findings: umbrella review. BMJ 361:k2387. Schmidt CO, Hegenscheid K, Erdmann P, Kohlmann T, Langanke M, Volzke H, Puls R, Assel H, Biffar R, Grabe HJ. 2013. Psychosocial consequences and severity of disclosed incidental findings from whole-body MRI in a general population study. Eur Radiol 23(5):1343-51
--	--

VERSION 1 – AUTHOR RESPONSE

Reviewer: 1
AJ Larner

Information on the prevalence of incidental findings on brain imaging and blood tests in older people is very welcome, not least because it is highly likely that increasing numbers will undergo these investigations as the population ages.

Although this is a research study, with clearly defined protocols, the results have potential application to inform clinical practice. Chief among these may be pre-test counselling of patients on the risks of possible incidental test findings, a particular issue with brain imaging which may be requested by clinicians with little or no experience of image interpretation.

We thank Dr Larner for these kind comments

Understandably the study has focussed on potentially serious imaging findings, but in clinical practice the finding of incidental white matter hyperintensities (a “non-reportable” finding here) often generates anxiety, sometimes being interpreted by non-neurological clinicians as indicative of “stroke” or even “vascular dementia”. It would have been useful from a clinical standpoint to have included data on the prevalence of this incidental finding. I would anticipate that such changes were seen in a very high percentage of the Insight 46 cohort.

The focus of this study was on potentially serious imaging findings. We have, however, undertaken separate analyses of white matter burden (Lane et al, Lancet Neurology, In Press). Its distribution is highly non-linear and so it is difficult to determine a threshold of abnormality. We have mentioned this in the discussion, referencing the (in press) paper.

Potential shortcomings of the study relate to the generalizability of the findings. This was a research population selected by date of birth, one week in 1946, and location of birth, mainland Britain. No information on patient ethnicity is given but it might be anticipated that the cohort would include few participants of either Afro-Caribbean (pre-Windrush) or Asian ethnicity. The findings might not, therefore, be entirely representative of the current British population aged 70. The necessity of travel to London for the study investigations might mean that, although participants were drawn from throughout mainland Britain, those from England might be over-represented compared to those from Scotland and Wales (no details of patients’ geographical origin are given).

These points about ethnicity and generalisability are all valid and are now mentioned in the revised discussion, with reference to our paper published on the representativeness of study (James et al, BMC Research Notes, 2018)

Minor points for correction:

p10, line 56: “data was” to read “data were”.

Reference 17: “Grimely EJ” to read “Grimley Evans J”.

References 21 and 22: only 2 authors given before et al., unlike previous references.

These have all been corrected

Reviewer: 2

Reviewer Name: Lorna M. Gibson

Thank you for the opportunity to review this interesting manuscript. Keuss et al. present the incidental findings detected on brain magnetic resonance imaging (MRI) and blood tests during the first phase of Insight 46, a longitudinal prospective sub-study of the 1946 British Birth Cohort. Empirical data on the frequency and types of incidental findings detected in healthy volunteers are extremely useful to practically inform future incidental findings feedback policies and the design of participant consent materials. My recent work has involved a systematic review of the prevalence and types of incidental findings on brain (as well as body) MRI, and a study of the impact of feedback of potentially serious incidental findings in the UK Biobank study, and I remain a member of UK Biobank's incidental findings team. This perspective clearly informs the comments I have made below, and which I hope are of use to the authors. Their work is interesting and relevant, but could be clarified in places and make much more use of the experience and insight of other pertinent studies. I have numbered my comments for ease of response.

We thank Dr Gibson for these helpful comments

1. The first bullet point of the strengths and limitations paragraph states that 'A large number of participants underwent blood testing and brain imaging, at an almost identical age, according to a pre-specified standardised protocol,' but research studies are usually conducted to a pre-specified standardised protocol. Do the authors mean to instead say that 'A large number of participants underwent blood testing and brain imaging, at an almost identical age, and received feedback of potentially serious incidental findings according to a pre-specified standardised protocol?'

Yes – this has been corrected

2. On page 6/32, the statement 'There are, however, important ethical reasons for disclosing certain incidental findings to participants in appropriate circumstances, particularly when they relate to serious and potentially treatable conditions' could be enhanced by a further reference to Wolf et al. 2008 (listed as reference number 1).

We have added this reference

3. On page 6/32, the statement 'Incidental findings often lead to anxiety and have the potential to lead to unnecessary and invasive procedures for study participants' requires referencing as these outcomes have been demonstrated by teams from the UK Biobank (for transparency, I am a co-author of this study), the German National Cohort and the Netherlands Epidemiology of Obesity Study (de Boer et al., 2018; Gibson et al., 2018a; Schmidt et al., 2013).

We have added these references

4. On page 7/32, the statement 'Several studies have reported rates of incidental findings in different samples previously,[5-10]' could be brought up to date with references to two recent systematic reviews (again, for transparency, I am a co-author of one of these studies) (Gibson et al., 2018b; O'Sullivan et al., 2018), rather than selected studies included in those two reviews.

We have added these references

5. On page 8/32 within the 'recruitment' subsection, can the authors provide more detail on what constituted 'relevant life course data' to allow readers to judge whether or not this selection criterion may have introduced bias.

We have included this information as a Supplementary File.

6. On page 8/32, the 'consent' subsection could be expanded to provide some important detail which would inform current debates on ethical feedback policies. In particular, please could the authors provide: a link, reference or supplementary file containing the study consent form and a short description or quote in this section on the wording of the consent form with regard to the participants' consent to receiving feedback of incidental findings; the wording for the option to 'opt out' of receiving correspondence about blood test results (and presumably the lack of option to 'opt out' of receiving correspondence about brain MRI incidental findings?); justification for the offering of this 'opt-out' option; details on whether or not participants were made aware that a lack of feedback did not constitute a health check, given the limitations of the research imaging for providing firm clinical diagnoses.

We have included the consent forms as a supplementary file. We have also expanded on this section of the text to describe how the study protocol regarding incidental findings was explained to participants in the information booklet. We gave participants the option of opting out of routinely receiving a copy of their blood results, primarily to prevent them from being overwhelmed, with a view to writing to these individuals only if there were actionable findings. However, in practice, the vast majority opted to receive a copy of their results.

7. On page 8/32, the final line should read 'MRI data were acquired simultaneously.'

Corrected

8. On page 9/32, the first use of 'FLAIR' needs to be defined in full.

Added

9. On page 9/32, the 'duty of care protocol for neuroimaging' section could also be expanded. Please could the authors provide details on (or a reference to) how they arrived at the justification for consultant neuroradiologist review of all images for their study, as later on page 18 they state that 'Most imaging studies, for example, do not require routine review of all scans by a radiologist.[13]'

Given that participants were all scanned at a single centre with the availability of consultant neuroradiologists, and due to the unique nature of the cohort, it was decided that all scans would have a radiologist review. We have explained this in the text.

10. The list of reportable and non-reportable findings has been adapted from the UK Biobank study lists (which were originally adapted from work by the German National Cohort),(Bertheau et al., 2016) but the size cut-off for feedback of intracranial aneurysms appears to have come from the Rotterdam Scan Study protocols (> 7 mm) (Bos et al., 2016). These three studies should be referenced in the text and in a footnote to Table 1.

These references have been added

11. On page 10/32, I would like to see more detail on the radiologists' reporting processes. Specifically, did the 'radiological read report' consist of the findings listed in Table 1 being marked as present or absent (i.e. reporting to a pre-specified checklist)? In addition, please provide details on the clinical information provided to the radiologists to aid the interpretation of images (e.g. age, sex, clinical history, previous imaging), as this will likely influence the prevalence of incidental findings.

We have added a Supplementary File with the radiological read completed by radiologists on XNAT. The radiologists were not given any clinical information when reporting scans, beyond knowing that participants were all born in 1946. We have clarified this in the text.

12. On page 10/32, the second to last line should read 'and then recommended clinical action.'

Not amended – meant to read as is.

13. On page 10/32, the authors state that 'since data was [sic] collected in an anonymised form, it was not possible to share the images for clinical use.' This is in contrast to UK Biobank (GPs can request copies of images for participants with potentially serious incidental findings), and could the authors please explain this further.

MR images were collected using a unique participant number. The only place that this unique number is linked to identifiable participant details is on a spreadsheet stored in a secured data safe haven. Sharing images with other health care professionals would have meant compromising the confidentiality of participants and their data. In addition, images do not meet standards for clinical use, which requires at least three identifiers (e.g. name, date of birth, hospital number).

14. On page 11/32, could the authors clarify whether blood test results outside the normal reference range were fed back only to participants, or to participants and their GPs?

They were fed back to both participants and their GPs. This is now made clear in the text.

15. On page 12/32, before the analysis section, can the authors insert a sentence to state whether or not participants were systematically followed up, as the results section is inconsistent in reporting findings as 'suspected intracranial mass lesions' versus 'intracranial mass lesions,' and it is not clear if the latter description reflects the initial incidental finding appearance, or a final diagnosis. During our systematic review of studies of potentially serious findings on brain (and body) MRI, we found limited reporting of firm final diagnoses generated after systematic follow-up, and during our study of UK Biobank participants, around 80% of findings detected by radiologists turned out not to be serious, i.e. many descriptions of incidental findings changed substantially after follow-up. Throughout this current manuscript it is important to clearly distinguish between descriptions of the incidental finding (i.e. a suspected condition demonstrated on non-diagnostic imaging), versus a final diagnosis of disease; this would set this work apart from other studies in terms of clarity of reporting.

We did not systematically follow up participants with regards to incidental findings detected. We have, however, obtained information on outcomes via different sources. We have now summarised this in a detailed Supplementary File and referred to it in the text.

16. On page 12/32, the authors need to add: their methods for calculating 95% confidence intervals (e.g. an exact method which assumes a binomial distribution, given the data are of prevalence); whether they conducted one- or two-tailed significance tests; the p-value they considered significant; how they dealt with missing data (e.g. the reduced denominators detailed in Table 4, and the participants who did not complete imaging (6.2%)).

We have re-calculated the confidence intervals using the exact Clopper-Pearson method, and added that we used a two-tailed significance test with p-value <0.05 considered significant. We have also clarified how we dealt with missing data.

17. On page 12/32, the authors describe the participants' ages as '70.7 (+/-0.7).' Is this the standard deviation? Please clarify.

Yes, it is – we have made this clear in the text.

18. Can the authors provide details (in a supplement perhaps) of the 15 findings initially flagged by the neuroradiologists and deemed not for feedback (i.e. 42% of the findings flagged by the consultants) and add a sentence to the discussion to describe why these did not (or why the 58% did) make it through the committee decision. Do the authors have any reflections on the study's incidental findings process as a result of this, given the large proportion of consultant-flagged findings which were not eventually fed back?

We have listed these findings in a Supplementary File. In all cases, the abnormality was not reportable as per the study protocol. We have mentioned this in the discussion.

19. On page 13/32, please state how many participants received feedback of a reportable finding from either brain MRI or blood tests, in order to clearly describe how many people were affected by feedback (of any type) during Insight 46. Also please specify how many participants had more than one reportable finding (from either brain, blood tests, or across both tests), again to illustrate the burden on participants.

We have now stated how many participants had a reportable finding on either blood tests or brain imaging, and how many participants had more than one finding.

20. Our systematic review did not find any convincing evidence of a difference in prevalence of potentially serious incidental findings on brain and body MRI between men and women (Gibson et al., 2018b). Please add one or two sentences in your discussion about your findings of differences in prevalence of reportable incidental findings on both the brain MRI and the blood tests between men and women.

We now comment on sex differences in imaging findings in the discussion. On reviewing the blood result data, we discovered that removing 'low creatinine' as an abnormality resulted in there being no significant difference between men and women. We have added this to the results section.

21. On page 13/32, please clarify if the most common vascular abnormalities were cerebral aneurysms, or suspected (or likely?) cerebral aneurysms, and if the most common intracranial lesions were meningiomas or suspected meningiomas, to clearly distinguish between a description of an incidental finding and of a final diagnosis.

We have clarified this in the text.

22. It is not clear whether or not (selected or unselected groups of) participants were followed up after they received feedback, how this was done, and over what time period. Please can you clarify in this in your methods section, as the final paragraph on page 13/32 (which extends on to page 14/32) and in the section on results of blood tests, you provide considerable detail on the outcomes of what appears to be a selected group of participants. We have recently documented that few studies seem to systematically follow up participants who receive feedback of incidental findings (Gibson et al., 2018b), and if the Insight 46 study did indeed conduct this, the methods and results should be much clearer as this would be a valuable contribution to the field. If however, only selected groups of participants were followed up, and the text reflects some of the more interesting cases, please be clear about this and state in the discussion that the data do not accurately reflect the burden of follow-up experienced by the entire group, nor do they allow for quantification of the numbers and types of

follow-up and that more data are clearly needed on this to inform judgements on the benefits and harms of feedback of incidental findings.

We have addressed this under point no. 15 and now reflect on this in the discussion.

23. Your finding that few participants opt out of feedback of incidental findings on blood tests is similar to the finding from the German National Cohort (Hegenscheid et al., 2013) that hardly any participants opted out of feedback of imaging findings, and a reference should be included in the 'comparison with other studies' section of the discussion.

We have added this reference.

24. Please add a short footnote to table 4 to explain the variability in denominators and how the missing data were dealt with so that this table can stand alone independently from the text.

This has been added.

25. The headings called 'Number' in Table 4 would be more accurately labelled 'n/N.'

We have amended this.

26. I do feel that the 'Comparison with other studies' section should begin with references to the two recent large systematic reviews of prevalence of incidental findings on brain MRI I mentioned above (Gibson et al., 2018b; O'Sullivan et al., 2018) rather than a selection of five studies contained within these. You are correct in stating that the prevalence reported in our review is lower than in the current Insight 46 cohort, although O'Sullivan et al. found far higher prevalences and you may wish to discuss your work in light of their result too.

We now reference these two systematic reviews and discuss them in the text.

27. You state that 'Most imaging studies, for example, do not require routine review of all scans by a radiologist.[13] Often, researchers will only ask for a radiologist opinion if an abnormality is identified incidentally by a radiographer during scanning or by researchers during data analysis.' In contrast to your statement, the vast majority of studies included in our review used at least one radiologist reader, and you do not provide any references to support your second statement. While the UK Biobank does employ a process of radiographer flagging of concerning images for a radiologist to review (Gibson et al., 2018a), the German National Cohort train radiologists to read all their imaging (Bertheau et al., 2016), and the Rotterdam Scan Study train readers to review all images of which those scans with concerning findings are then reviewed by two neuroradiologists. I think your two sentences quoted above are worth clarifying and referencing, as the protocols for incidental findings vary considerably (as is appropriate), depending on the context of the study.

We have added a further reference to support our statements. We also acknowledge, with reference to your review, that many studies do involve radiologist review. However, the fact that most studies in your review used at least one radiologist reader may reflect that such studies are more likely to publish data on prevalence of incidental findings than those that do not involve some sort of radiological review. This is also mentioned in the text.

28. The prevalence estimates of incidental findings detected during the Rotterdam Scan Study (reference number 6), which you describe on page 19/32, have been updated (Bos et al., 2016) and your text should reflect this.

This has been updated.

29. In the 'strengths and weaknesses' subsection on page 20/32, I think again the authors mean to say 'A large number of participants underwent blood testing and brain imaging, at an almost identical age, and received feedback of potentially serious incidental findings according to a pre-specified standardised protocol.'

Yes – this has been amended.

30. On page 19/32, please change 'images were systematically reviewed by two experienced consultant neuroradiologists' to 'images were systematically reviewed by one of two experienced consultant neuroradiologists' so it is consistent with your methods section.

Corrected

31. On page 19/32 you describe your feedback process as 'in keeping with the ethical principle of beneficence.' It can be argued that participants in research studies do not stand to benefit from participating, nor is there convincing evidence of benefit (although there is evidence of real harm) of feeding back incidental findings (Gibson et al., 2018a). Indeed, the research imaging performed in your study is not a 'health check.' Please expand on your reasoning here.

We entirely agree that research studies differ from systematic health checks. However, despite this being made explicit in study protocols, many participants do view medical input as a benefit of taking part in research. We mention this in the text, also acknowledging the potential negative consequences of reporting back incidental findings

32. On page 21/32 you mention that 'participant perception regarding the disclosure of incidental findings was not formally assessed, nor was the impact on their longer-term health and psychological wellbeing.' Please add text here to say that data on final diagnoses were not systematically collected (if this is the case). Also, if there are plans to assess these outcomes in the Insight 46 cohort please do describe this here, as this would be very useful to inform debates of the benefits and harms of feedback of incidental findings.

We have addressed this under point 15, and now mention this in the future work section of the discussion.

33. Your statement on page 21/32 ('Many participants, however, gave informal feedback on post-visit evaluation forms that they appreciated being told about findings pertinent to their health and saw this as a benefit of being involved in the study') is consistent with the findings of the UK Biobank, the German National Cohort, and the Netherlands Epidemiology of Obesity Study and these would be worth referencing (de Boer et al., 2018; Gibson et al., 2018a; Hegedus et al., 2019).

We have added these references.

34. On page 21/32, you state 'Moreover, almost all participants chose to receive a copy of their blood test results. These observations are consistent with results of a study commissioned by the Wellcome Trust and MRC, which found overwhelming public support for the disclosure of incidental findings in research, particularly in relation to serious and treatable conditions.[20]' But you state in your methods

'Results of the clinical blood tests were reviewed by the study doctor and reported back to the participant and their GP in writing within two weeks of the study visit.' My reading of this is that everyone received feedback of all blood test results. If that is the case, then I wonder instead if your findings reflect participants' wishes to receive any feedback, regardless of seriousness (which would be need references to different studies which demonstrate this (Kirschen et al., 2006)). Please clarify.

We have clarified this by amending the sentence in the methods section to say that blood test results were reported back to participants if they had opted to receive a copy, or if there was something actionable.

35. In the 'implications and future work' section, you state that 'By defining what is actionable and providing the expected prevalence of incidental findings on brain MRI and clinical blood tests in this age group, researchers may be better prepared for managing incidental findings, and participants [sic] better informed of their likelihood as part of the consent process.' However the lists of findings used by Insight 46 are based extremely closely on the work of other studies (initially an extensive piece of work by the German National Cohort, which was then adapted by UK Biobank, which I think you have further adapted and combined with the Rotterdam Scan Study policy). In light of this I do not think you can claim that Insight 46 has defined what is actionable in terms of a finding for feedback. Instead, you can certainly say that Insight 46 has demonstrated that the work of these other studies (predominantly the German National Cohort, to give credit where it is due) appears to be adaptable and feasible to use in other imaging contexts.

We have re-worded this section of the discussion. We did not mean that we defined what was actionable (we already acknowledge in the methods section that the lists used were adapted from other studies). Instead, what we meant is that using pre-defined lists of reportable findings and being aware of the expected prevalence of incidental findings should allow researchers to be better prepared for managing them.

36. Your closing statement describes work which continues to be conducted by several large European studies I have already mentioned above (among others), some of which is already published (Bos et al., 2016; de Boer et al., 2018; Gibson et al., 2018a; Hegedus et al., 2019; Hegenscheid et al., 2013; Schmidt et al., 2013), and you should reference them. A stronger closing statement may describe how your current manuscript also feeds in to the debate on ethical feedback policies and design of the consent materials (specifically those used in the Insight 46 study, or perhaps birth cohorts more generally).

We have now referenced these studies, and added a stronger closing statement, reflecting on how this study might contribute to current research in this field.

References

Bertheau RC, von Stackelberg O, Weckbach S, Kauczor H-U, Schlett CL. 2016. Management of incidental findings in the German National Cohort. In: Weckbach S, editor. *Incidental radiological findings*. First ed. Cham, Switzerland: Springer.

Bos D, Poels MM, Adams HH, Akoudad S, Cremers LG, Zonneveld HI, Hoogendam YY, Verhaaren BF, Verlinden VJ, Verbruggen JG, Peymani A, Hofman A, Krestin GP, Vincent AJ, Feelders RA, Koudstaal PJ, van der Lugt A, Ikram MA, Vernooij MW. 2016. Prevalence, clinical management, and natural course of incidental findings on brain MR images: the population-based Rotterdam Scan Study. *Radiology* 281(2):507-515.

de Boer AW, Drewes YM, de Mutsert R, Numans ME, den Heijer M, Dekkers OM, de Roos A, Lamb HJ, Blom JW, Reis R. 2018. Incidental findings in research: a focus group study about the perspective of the research participant. *J Magn Reson Imaging* 47(1):230-237.

Gibson LM, Littlejohns TJ, Adamska L, Garratt S, Doherty N, UK Biobank Imaging Working Group, Wardlaw JM, Maskell G, Parker M, R. B, Matthews PM, Collins R, Allen NE, Sellors J, Sudlow CLM. 2018a. Impact of detecting potentially serious incidental findings during multi-modal imaging [version 3; referees: 2 approved, 1 approved with reservations]. *Wellcome Open Res* 2:114.

Gibson LM, Paul L, Chappell FM, Macleod M, Whiteley WN, Salman RA, Wardlaw JM, Sudlow CLM. 2018b. Potentially serious incidental findings on brain and body magnetic resonance imaging of apparently asymptomatic adults: systematic review and meta-analysis. *BMJ* 363:k4577.

Hegedus P, von Stackelberg O, Neumann C, Selder S, Werner N, Erdmann P, Granitza A, Volzke H, Bamberg F, Kaaks R, Bertheau RC, Kauczor HU, Schlett CL, Weckbach S. 2019. How to report incidental findings from population whole-body MRI: view of participants of the German National Cohort. *Eur Radiol*. (EPub ahead of print)

Hegenscheid K, Seipel R, Schmidt CO, Volzke H, Kuhn JP, Biffar R, Kroemer HK, Hosten N, Puls R. 2013. Potentially relevant incidental findings on research whole-body MRI in the general adult population: frequencies and management. *Eur Radiol* 23(3):816-826.

Kirschen MP, Jaworska A, Illes J. 2006. Subjects' expectations in neuroimaging research. *J Magn Reson Imaging* 23(2):205-9.

O'Sullivan JW, Muntinga T, Grigg S, Ioannidis JPA. 2018. Prevalence and outcomes of incidental imaging findings: umbrella review. *BMJ* 361:k2387.

Schmidt CO, Hegenscheid K, Erdmann P, Kohlmann T, Langanke M, Volzke H, Puls R, Assel H, Biffar R, Grabe HJ. 2013. Psychosocial consequences and severity of disclosed incidental findings from whole-body MRI in a general population study. *Eur Radiol* 23(5):1343-51.

VERSION 2 – REVIEW

REVIEWER	Dr Lorna M. Gibson Usher Institute of Population Health Sciences and Informatics, University of Edinburgh, Scotland, UK Member of the UK Biobank Imaging Working Group. UK Biobank Imaging Consultant.
REVIEW RETURNED	21-Jun-2019
GENERAL COMMENTS	Many thanks indeed for addressing all of my previous comments so comprehensively. I have no further comments or questions, and I support the publication of this manuscript.